# A wind-albedo-wind feedback driven by landscape evolution

Jordan T. Abell [1,2]*, Alex Pullen[3], Zachary J. Lebo [4], Paul Kapp [5], Lucas Gloege [1,2], Andrew R. Metcalf [3], Junsheng Nie [6] & Gisela Winckler [1,2]

The accurate characterization of near-surface winds is critical to our understanding of past and modern climate. Dust lofted by these winds has the potential to modify surface and atmospheric conditions as well as ocean biogeochemistry. Stony deserts, low dust emitting regions today, represent expansive areas where variations in surficial geology through time may drastically impact near-surface conditions. Here we use the Weather Research and Forecasting (WRF) model over the western Gobi Desert to demonstrate a previously undocumented process between wind-driven landscape evolution and boundary layer conditions. Our results show that altered surficial thermal properties through winnowing of fine-grained sediments and formation of low-albedo gravel-mantled surfaces leads to an increase in near-surface winds by up to 25%; paradoxically, wind erosion results in faster winds regionally. This wind-albedo-wind feedback also leads to an increase in the frequency of hours spent at higher wind speeds, which has implications for dust emission potential.

[1] Lamont-Doherty Earth Observatory, Columbia University, Palisades, NY, USA. [2] Department of Earth and Environmental Sciences, Columbia University, New York, NY, USA. [3] Department of Environmental Engineering and Earth Sciences, Clemson University, Clemson, SC, USA. [4] Department of Atmospheric Science, University of Wyoming, Laramie, WY, USA. [5] Department of Geosciences, University of Arizona, Tucson, AZ, USA. [6] Key Laboratory of Western China's Environmental Systems (Ministry of Education), College of Earth and Environmental Sciences, Lanzhou University, Lanzhou 730000, China. *email: jabell@ldeo.columbia.edu

Changes to surface albedo, sensible heating, wind, and aerosols result in feedbacks within the Earth's climate[1–3]. Nonlinear feedbacks, defined here as a system of two or more variables that do not have a linear effect on one another, are of primary concern for climate modeling because they are responsible for a large contribution to uncertainties[3–5]. We present a previously undocumented feedback between near-surface winds and landscape evolution that, if incorporated into climate models, may prove useful in improving the accuracy to which we can simulate past and future climates.

Climate simulations aiming to characterize ocean and atmospheric circulation as an integrated system allow land-based ice volumes, vegetation, and surface moisture to fluctuate; few accurately depict landscape evolution—the change in alluvium and barren bedrock exposed at the surface—which is fundamentally important for desert regions lacking ice, substantial vegetation or abundant moisture. The use of (near-) modern surficial geological parameters (e.g., erodibility, albedo) for arid basins may contribute to errors in wind speeds and subsequent dust production in climate simulations.

The arid basins within the East Asian continental interior experience high near-surface wind speeds and are the Earth's second largest sources of lithogenic aerosols[6–8]. The East Asian dust-producing regions are upwind of a prominent high-nutrient/low-chlorophyll area in the North Pacific where the productivity of photosynthetic organisms is limited by the concentration of bioavailable Fe within surface waters[9–11]. To those ends, understanding winds and dust production in East Asia has significant implications[12].

Stony surfaces comprise >50% of the Earth's deserts, but account for a much lower percentage of dust emissions[13–15]. Specifically, stony deserts compose >70% of the total desert area of East Asia, but only account for ~30% of total dust emissions[16] (Supplementary Fig. 1). Development of these low dust-emitting desert pavements can occur by processes such as heaving, deflation of fine-grained material, aggradation of fine-grained material causing inflation of existing pavement, among others[17–20]. Once formed, tightly spaced stones armor the surface and suppress further wind deflation[17–24]. Coalescence of (usually dark-colored) stony surfaces occurs on $10^4$–$10^6$-year timescales and alters the thermal properties of desert surface by lowering albedos and increasing sensible heating[25].

The Gobi Desert, China provides a natural laboratory to test the relationship between wind deflation, darkening of the land surface, and wind speeds (Fig. 1a). The Turpan-Hami depression is hyperarid (<100 mm yr$^{-1}$ of rainfall) and can be used to test this relationship because it has >100 days yr$^{-1}$ with 10-min averaged gusts of >17.2 m s$^{-1}$ and a mean annual speed of ~8 m s$^{-1}$ [26,27]. These high wind speeds allow us to isolate the landscape driven wind-albedo affect in a regional climate model. Today, over ~70% of the Turpan-Hami surface consists of dark gray, low-albedo (~10–15%)[28,29] unconsolidated eolian gravel lags[30], which are partially responsible for the high mountain to basin temperature gradient and act as a type of pavement (Fig. 1a). The gravel lags coalesced over the past ~3 million years from wind deflation of interbedded alluvial fan (conglomerate) and playa (mudstone/siltstone) strata, which would have had a much higher surface albedo prior to wind erosion when only the rock is considered (Fig. 1b)[30]. Wind erosion has removed an integrated average of at least 180 m of sedimentary bedrock over the past ~3 million years (Fig. 1a, f)[30]. Landscape evolution has transformed the albedo, and thus the thermal properties of the surface (Fig. 1c, e). The change to the land surface through time likely had effects on the properties of the atmospheric boundary layer, particularly near-surface wind speeds, which would continue to erode and shape the landscape.

Here, we conduct Weather Research and Forecasting (WRF) modeling experiments on the Turpan-Hami depression and surrounding areas for the spring months of March, April, and May (MAM), 2011, to identify the consequences of long-term, continuous wind erosion on near-surface winds and other components of the planetary boundary layer. Only the background surface albedo is changed in order to isolate the wind-albedo effect, and the names of the simulations correspond to the percent albedo used for the Hami Basin area (Supplementary Table 1; see the Methods section for model setup). Parameterization the albedo is from the U.S. Geological Survey (USGS)[28,29], which is then compared with adjacent sandy deserts in Asia (e.g., Taklamakan: 25–35%; Qaidam Desert: 30%) and the major desert regions of the Sahara and Arabian Peninsula (40–45%) to ground truth[31]. Our results indicate that wind erosion and the subsequent production of gravel-dominated surfaces leads to an increase in near-surface winds speeds by up to 25%, as well as increases in surface vertical wind speeds, surface turbulent kinetic energy, and an approximate doubling of the average height of the planetary boundary layer over the spring season. In addition, the frequency of hours spent at high (potentially dust-deflating) wind speeds also increases by ~30–40%.

## Results

**Model validation.** Considerable efforts to validate the control simulation (WRF_Control) are taken to demonstrate that the WRF model accurately simulates climate conditions during the simulated period over the western Gobi region, including comparison with the forcing data set (NCEP Global Forecast System Analysis (GFS-ANL)), an independent reanalysis product (European Centre for Medium-Range Forecast's (ECMWF) fifth-generation atmospheric reanalysis (ERA5))[32], and daily observations at 14 surface point stations using the Integrated Surface Hourly data set collected by the United States Air Force (USAF) Climatology Center[33] (see Supplementary Note 1 for details).

For the comparison with reanalysis data sets, monthly means of 2-m temperature (T2), surface pressure (SP), 500-mb horizontal winds (500-Winds), and 10-m winds (10-Winds) are used. WRF_Control output for Domain 1 (D01) (Supplementary Fig. 2) is able to reproduce spring T2, SP, and 500-Winds very well, with root-mean-squared errors (RMSE) only reaching a maximum (between both data sets) of 2.5 K, 7.8 hPa, and 1.3 m s$^{-1}$, respectively (Supplementary Table 2; Supplementary Figs. 3, 4). For 10-Winds, there is a larger discrepancy between WRF_Control and the ERA5 and GFS reanalysis data sets, with the maximum monthly RMSE reaching 2.8 m s$^{-1}$ (Supplementary Table 2; Supplementary Fig. 4).

When considering local, ground-based observations, WRF_Control replicated the observed T2 and SP to a high degree (Supplementary Table 3). In addition, the WRF_Control simulation was able to capture >30% of the variance in the daily-averaged 10-Winds through the spring (Supplementary Table 3; Supplementary Fig. 5) for 8 of the 12 sites where p-values indicate a <5% chance that no relationship exists. Of those eight sites, six predominantly showed lower observed 10-Winds than those calculated in the model. For seven of the eight sites where the $r^2$ was >0.30 between the model and observations, the sign of the springtime trend of 10-Winds was accurately reproduced in the model (Supplementary Fig. 5).

Overall, the WRF_Control simulation satisfactorily reproduces the large-scale surface and synoptic meteorology in the western Gobi Desert, China. In addition, it succeeds in accurately representing the local conditions most relevant to this investigation, namely surface winds, surface pressure, and temperature at 2 m. Together, this provides confidence in using WRF to interrogate our hypotheses.

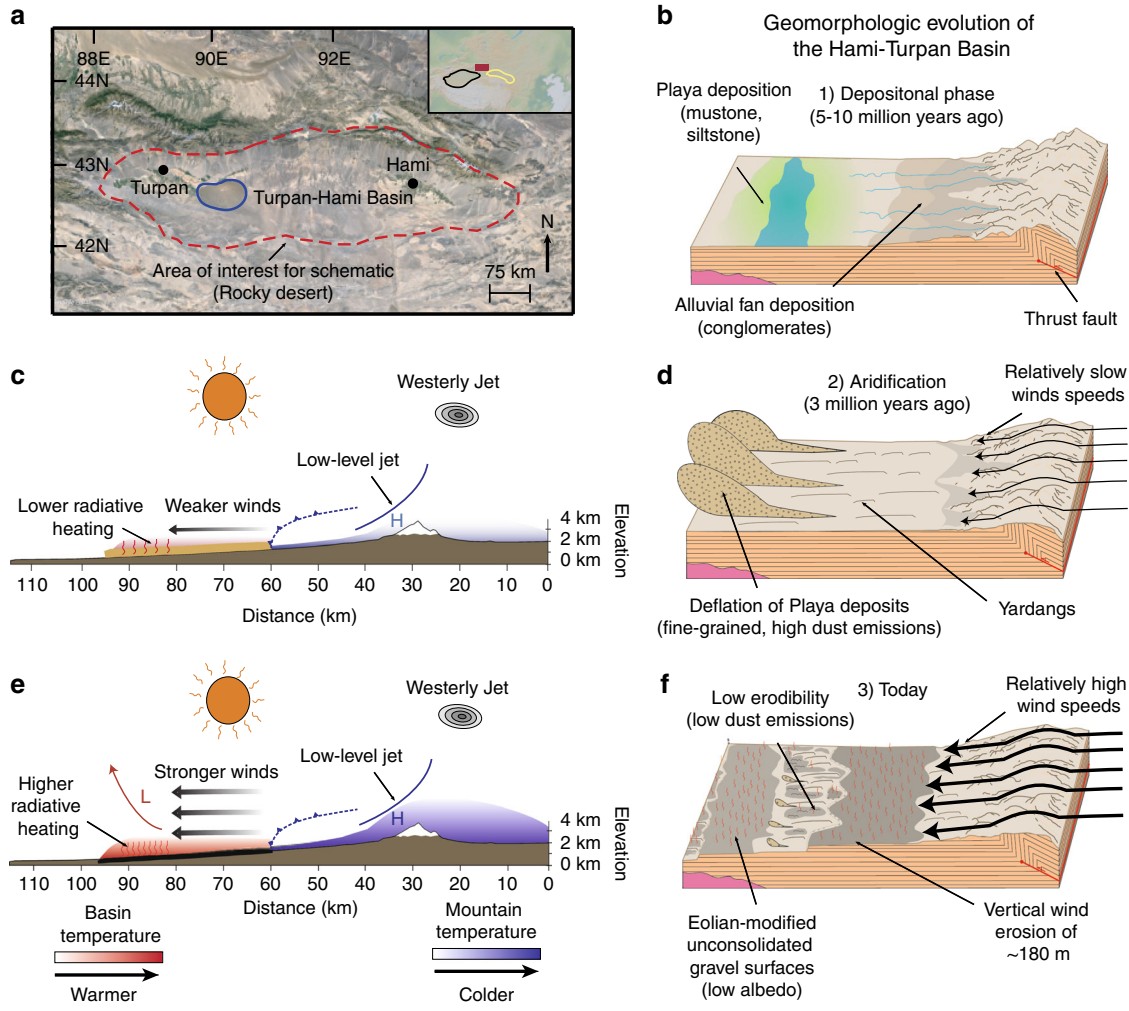

**Fig. 1 Schematic illustration of the geomorphologic development of the Hami Basin. a** Map of the Hami-Turpan depression, with major cities labeled. The red dotted line encircles the stony desert described by the schematic. Blue solid line indicates area of dune field. Inset displays physical map of East Asia. Red square represents the region shown on the larger map. Black and yellow lines mark the approximate extent of the Tarim Basin and Hexi Corridor, respectively. **b** Depositional phase of Neogene sediments. Conglomerate facies were likely derived from alluvial fans, while finer-grained siltstones and mudstones originated from playa deposition. **c, d** Regional climate system and landscape present during initial aridification of the Hami Basin, likely in the late Miocene or Pliocene. **d** The lighter-colored finer-grained playa deposits are easily wind-deflated, meaning more dust emissions. **c** However, northerly winds moving across the depression were likely weaker than present, due to a smaller mountain-basin temperature gradient from higher albedo of the low-lying basin. **e, f** Present-day version of the Hami landscape and climate. **f** Vast areas of dark-colored, coarse-grained topographically tiered, eolian-modified gravel surfaces, exposed by wind deflation of finer particles, maintain low albedo. **e** The lower albedo increases surface temperature of the depression relative to (**c, d**) and creates a higher temperature gradient, leading to higher surface wind speeds. Modified from ref. [30]. Figure 1a made with GeoMapApp (www.geomapapp.org)/CC BY/CC BY[69] and using Google Earth—US Department of State Geographer, ©2018 Google, Image Landsat/Copernicus.

**Albedo impacts on near-surface conditions**. To understand the effect of surface albedo changes on local and regional meteorology, we focus on three particular spatial/temporal combinations. These include variables averaged over the entire spring within Domain 1 (results only for Albedo_40) (Fig. 2a), averaged over the entire spring within Domain 3 (D03) (results only for Albedo_40) (Fig. 2B), and averaged monthly to a single value within the Hami Basin where the albedo was altered (all simulations considered) (Fig. 3a). Albedo_40 is selected as it represents the most likely scenario for a pre-desert pavement Hami Depression based on the bedrock geology. To provide more detail than what is presented here, statistical information for all variables of interest, and for all model runs, is shown in Supplementary Data 1.

For all simulations where the background albedo is greater than the control (modern) value (all values greater than ~12–15%), a decrease in T2 in the Turpan-Hami Basin is

observed. The decrease in temperature for a given increase in albedo follows a relationship found in urbanization model experiments[34]. The maximum temperature difference for the spring between the Albedo_40 and WRF_Control simulations within the Hami Basin is ~2 K, while no major changes are noticed elsewhere (Fig. 2a, b). Similar differences between the albedo simulations and the control run exist for the surface sensible heat flux (HFX) and the height of the planetary boundary layer (PBLH), with monthly means for Albedo_50 to Albedo_20 within the Hami Depression are ~10–75% and ~5–50% lower for these variables, respectively. For Albedo_40 specifically, the albedo difference corresponds to an average spring decrease of up to ~70 W m⁻² for HFX and ~500 m for PBLH. Again, no differences are observed outside of the region where albedo was altered.

Surface wind speeds and turbulent kinetic energy (TKE) in the Turpan-Hami Basin are also lower when albedo is higher than the

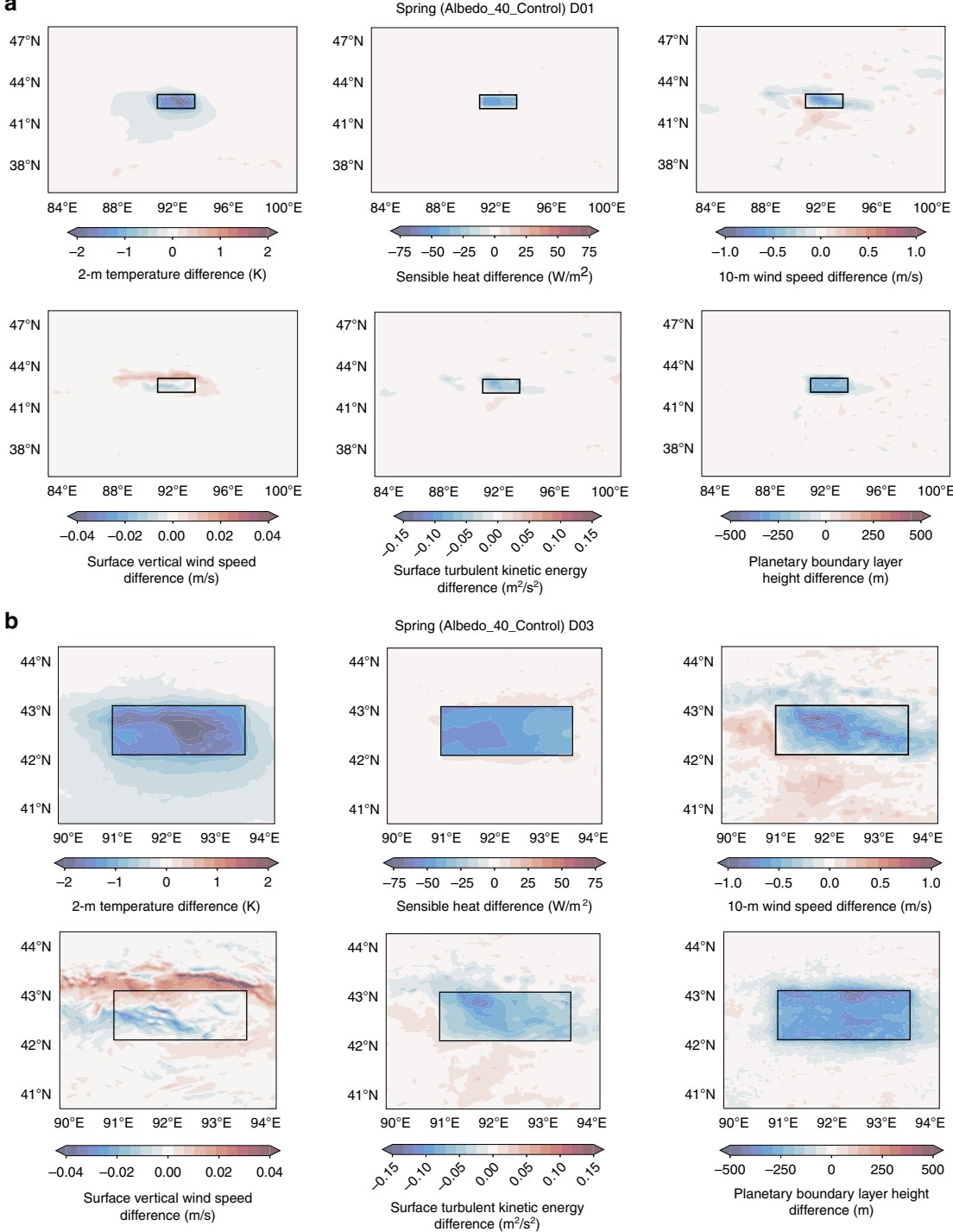

**Fig. 2 Spring means of various near-surface variables for the Hami Basin. a** Difference in spring (March, April, and May) means for 2-m temperature (T2), sensible heat flux (HFX), 10-m winds (Winds10), surface vertical wind speed (W), surface turbulent kinetic energy (TKE), and the height of the planetary boundary layer (PBLH) between the simulation where surface albedo was set to 40% (Albedo_40) and the control simulation (WRF_Control) across Domain 1. Averages were obtained from all 3-h time steps at each grid point. Blues indicate lower values in the Albedo_40 simulation compared with WRF_Control, while reds represent higher values. **b** Same as **a**, but for Domain 3.

control simulation, where the magnitude of this decrease increases with increasing albedo (Fig. 3b). Albedo_40 spring mean horizontal wind speeds (Winds10) show up to a ~1.5 m s$^{-1}$ reduction (corresponding to a range of <5% to ~25%) from the control (Fig. 2a, b). The maximum differences in 10-Winds varies through the spring months, but primarily peaks in April–May. The same is true for vertical surface winds speeds (W) and surface

TKE. Specifically, Albedo_40 average monthly differences from the WRF_Control run can reach ~30% for vertical wind speed and ~20% for TKE (Fig. 3b).

For 10-Winds, W, and TKE, there are notable differences between Albedo_40 and WRF_Control found outside the region where the albedo was changed, meaning there is evidence for not just a local effect but a regional one as well. Surface winds exit the

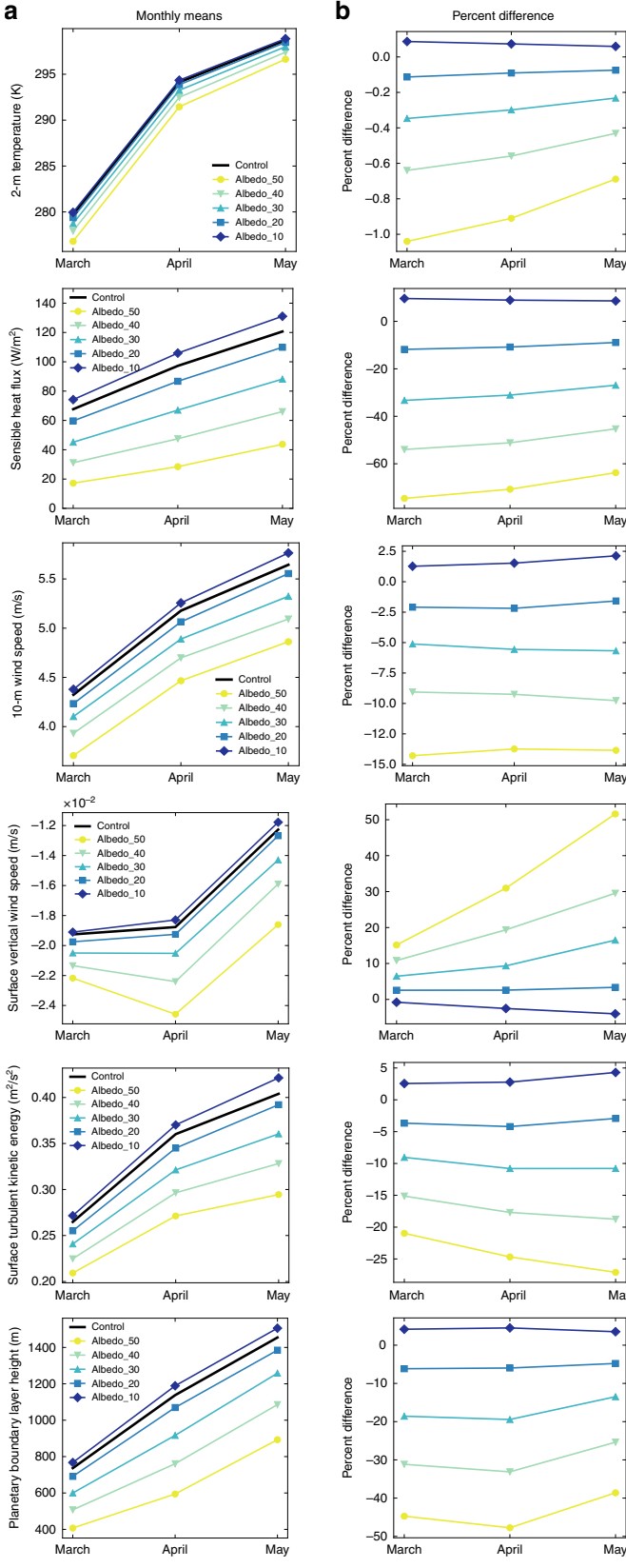

**Fig. 3 Monthly means and percent differences of various near-surface variables for the Hami Basin. a** Monthly means of corresponding variables for all albedo model runs (lines with markers) and the control run (thick black line) based on Domain 3 output. The values for each month are averaged for the region in which the albedo was altered (black box in 2**b**). **b** Percent difference from WRF_Control for each albedo simulation as compared with the control simulation. Negative values indicate a lower values in the Albedo_40 simulation compared with WRF_Control, while positive values indicate higher values.

Less significant (5–10%) decreases in 10-Winds are observed in the far eastern portion of the Hami, in the Turpan sub-basin, and on the slopes of the mountains bounding the region to the north (Fig. 2a, b). A dune field located between the Turpan and Hami sub-basins displays increases of 5–10% for 10-Winds (Fig. 2b), which is likely due to the decreased 10-Winds on the eastern and western sides of the dune field.

Due to the specific geologic setting of the Turpan-Hami region, vertical surface winds are difficult to interpret through simple difference calculations. Today (represented by WRF_Control), surface vertical winds move toward the surface in the mountain regions north of our study area, and in the northern Hami Basin. Opposingly, in the Hami itself, most of the vertical winds move upward due to high ground temperatures. Our Albedo_40 simulation shows decreases in the downward vertical wind speed within the mountain range to the north of the Hami Depression, and slower movement upward within the depression itself. This pattern is expected, as it likely results from weakened temperature gradients within our Albedo_40 simulation. Finally, TKE variations reflect changes in vertical and horizontal surface wind speeds in all areas mentioned previously.

Beyond just focusing on the mean of 10-Winds, the frequency of wind speeds over time is critical to understanding the impact of low-level winds on other parts of the earth system, specifically dust-emission potential. For Albedo_40, the number of hours during the spring of 2011 spent at speeds sufficient to deflate dust, assumed here to be ~5-6 m s$^{-1}$ [36,37] (the magnitude is dependent on roughness, vegetation, soil moisture, etc.) can range from ~25% higher to ~100% lower than WRF_Control (Figs. 4a, 5a). The typical range is between 0 and 40% lower. The range in wind speeds that show the largest reduction in frequency span 4–12 m s$^{-1}$ (Figs. 4b, 5b). The frequency of hours with gusts higher than this are quite low and similar between model simulations, and concomitantly, the higher albedo runs have more hours spent at lower (1–4 m s$^{-1}$) wind speeds (Fig. 4b).

## Discussion

The WRF simulations presented here provide quantitative evidence of a previously undocumented wind-albedo feedback mechanism. As the albedo is changed, the amount of absorbed incoming solar radiation is altered, which in turn changes sensible heating at the surface, altering temperature and pressure gradients across the basin. The physics behind the model results here are not novel[38–40], although the impacts for how we more accurately model paleo-landscapes, dust emissions, and related aerosol forcing are.

Because the stony surface of the Turpan-Hami Basin developed from eolian downcutting through sedimentary bedrock over $10^4$–$10^6$ years, the increase in mean wind speeds at lower modeled surface albedo supports the hypothesis that wind speeds increased as gravel lags coalesced[30] (Fig. 1). The relationship between surface albedo and wind speed has been previously documented and was to be expected (e.g. refs. [41–43]). Here, we emphasize the quantification of the net change in wind speed (e.g., up to 25% within the basin) and albedo's effect on other

Hami depression to the south and move (1) westward across the Hexi corridor and into the Tarim Basin/Taklimakan Desert or (2) eastward toward the deserts of East Asia[35] (Supplementary Fig. 4). There is an increase of <5 to ~15% in springtime 10-Winds across this region moving northeast/east to southwest/west (Fig. 2a, b).

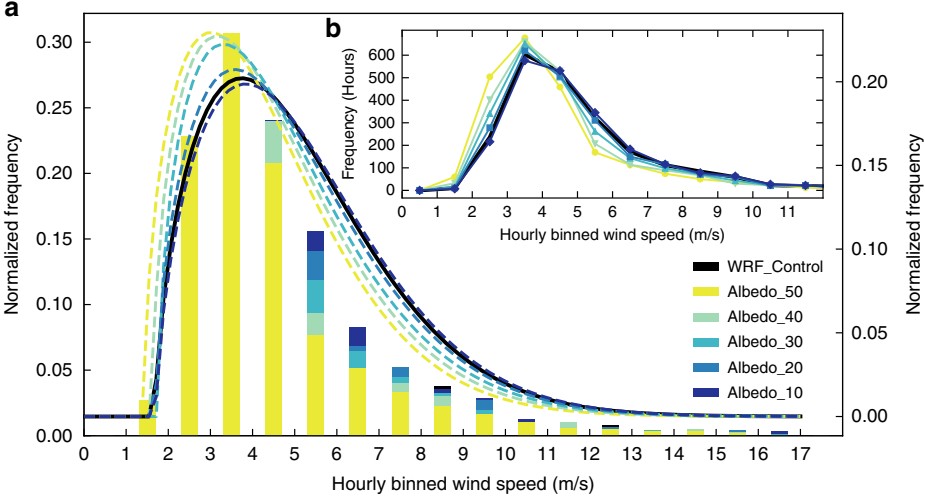

**Fig. 4 Histogram of surface horizontal wind speed frequencies for the spring months of 2011. a** Hours for spring (March, April, and May), 2011 in which the average 10-m horizontal wind speeds within the box where albedo was altered for Domain 3 were totaled and binned in 1 m s$^{-1}$ intervals to remove noise. Each bar represents the number of hours spent at (and including) the preceding value up to (but not including) the next wind speed. For example, the bars at 1.5 m s$^{-1}$ are all hours with average wind speeds between 1 and 2 m s$^{-1}$, but not including 2 m s$^{-1}$ or above. The control run is represented as black bars. The frequencies are normalized to 1 (left y-axis). Dotted colored and black lines are Weibull distribution fits for each of the model simulations, and are also normalized to 1 (right y-axis). Weibull fits were chosen due to the skewed nature of the 10-m horizontal wind speeds. **b** Same as **a**, but a line plot is used instead of bars to assist in differentiating between model runs. Frequency is now in total spring hours.

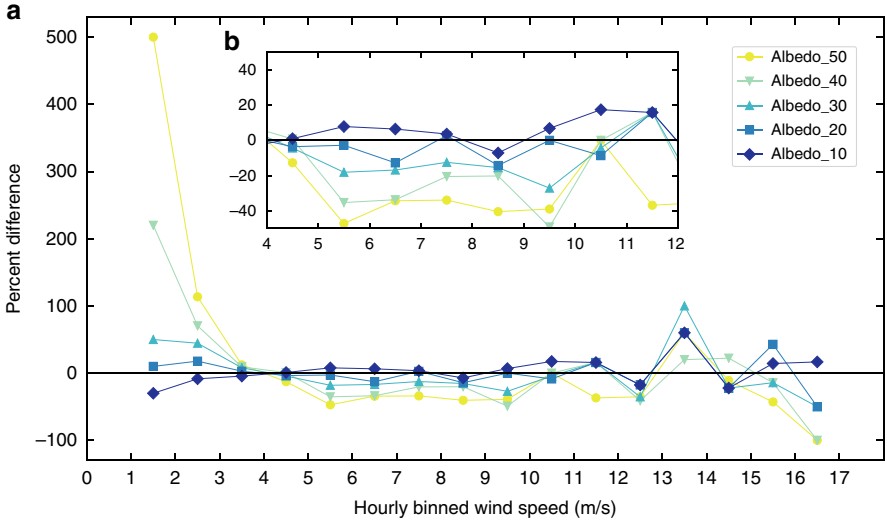

**Fig. 5 Percent difference of wind speed frequencies between the albedo simulations and control. a** The percent difference of spring (March, April, and May) 2011 hourly 10-m wind speed frequency (averaged for the entire Hami Basin) for each albedo simulation from the control simulation (WRF_Control). The binned data are from Fig. 4. **b** Same at **a**, but with reduced x- and y-axes limits to show small variation between model simulations.

surface and planetary boundary layer conditions, and a description of the wind-albedo feedback (i.e., albedo change via a process driven by wind erosion and resulting in faster winds). We note that the responses of the variables pertaining to the planetary boundary layers (TKE, W, and PBLH) are consistent with expectations for increased surface albedo and the subsequent decrease in near-surface winds.

While the relative difference in mean spring wind speeds between the higher albedo simulations and the control in the Hami Basin were substantial (Supplementary Data 1; Figs. 2, 3), we observe changes in the absolute values of the mean wind speeds and the frequency of high (potentially dust-deflating) wind speeds produced when the surface albedo is altered. At times, the change in mean wind speeds across sections of the Hami Basin go from subcritical velocities in Albedo_40 for the mobilization of particles to

supercritical velocities in WRF_Control, with additional dependence on the grain size and surface characteristics of the erodible materials. This dependence is impactful because winds, responding to the pressure gradient change from the decreased albedo (holding all else constant), became increasingly faster over time, which would have been necessary for continued wind deflation of surfaces on which gravels were coalescing. The increased frequency of high wind speeds when the albedo is lower (Fig. 4a, b) points toward increased gustiness and short periods of sustained high wind speeds. Gustiness has been shown to be a dominant driver of dustiness[44–48], as dust flux relates to the cube of the wind speed[49–51]. As such, the feedback involving wind, albedo, and increased frequency of high wind speeds likely played an even more important role in further landscape evolution and changes of dust emissions than that determined using only mean wind speeds and adds further credence

to the idea that this mechanism is important for describing the climatic impact of arid regions through time, on a global scale, and potentially in extraterrestrial settings[52].

In addition to the effects within the Hami Basin, regional consequences of the wind-albedo-wind feedback system are recognized in our study domain. Simulations with higher albedos prescribed within the Hami Depression show faster winds exiting the basin and traveling toward the Tarim Basin (Fig. 2a). This wind increase is likely caused by the alteration of temperature and pressure gradients between the Hami Basin itself and the region south of the basin. The increased winds are expected to lead to increased erosion of material, and likely increase dust production at the eastern margin of the Tarim Basin. As albedo decreased through time, this may have led to decreased erosion and dust emissions from an area upwind of the modern Hexi Corridor. While no larger, synoptic-scale climate effects are discussed here, it is possible that changes to even these regional-scale winds and their subsequent erosion patterns could have important implications downwind.

A full description of the wind-albedo-wind feedback allows for some consideration of its impact on dust emissions, as well as other factors that compound the system. The results of our experiments indicate that a decrease in albedo leads to an increase in mean wind speeds and the frequency of high wind speeds along with increased turbulent energy in the planetary boundary layer. Although no quantitative assessment of how these changes effect dust emissions is provided here, it is reasonable to suggest that all of these factors would lead to increased dust emissions through time from a purely atmospheric standpoint. However, modern observations show that dust deflation from the stony Gobi Desert is minimal when compared with other arid East Asian basins, despite the extreme gustiness and high mean annual winds. This indicates that the atmospheric changes are likely being offset by increasing mean grain size and surface characteristics, which act to decrease dust emissions[48]. Grain size increased as the gravel surface coalesced, and as grain size and dust emissions show a nonlinear relationship[48], this relationship confounds the question of dust flux through time. In addition, as gravel pavements develop, increased surface roughness can act to suppress dust emissions by extracting momentum from the surface winds[53–55]. Opposingly, as larger grains develop smoothed surfaces, it can also lead to decreased surface roughness[56], which can act to further increase wind speeds and thus dust emissions. Favoring dust suppression, the larger gravel clasts that cap the surface can trap finer grains beneath them, shielding them from further deflation[19]. Finally, to isolate a single variable and simplify the modeling experiment, we have assumed a homogenous albedo for each of our idealized simulations. However, homogeneity is not necessarily the case for two specific reasons. One, the gravel deposits cover ~70% of the basin floor[30], meaning the other 30% consists of lighter-colored deposits. In addition, some of these gravels are covered (to varying degrees) by desert varnish[30], which has darkened their surface. If older clasts are darker than those that are much younger, which lack varnish, then this would create a heterogenous albedo as well.

In summary, some of the processes described here are considered nonlinear and thus problematic for accurate climate modeling. This is where the paradox notion of the wind-albedo-wind feedback is derived from. Wind deflation led to the armoring of the land surface, which resulted in an increase in mean wind speeds and gustiness. This informs us that albedo, which is determined through surficial geology (in this case), can have a positive feedback on wind speeds, with the initial albedo perturbation driven by wind erosion. However, the potential change in dust emissions resulting from the increasing near-surface wind speeds and the suppression of dust emissions via shielding and increased grain sizes brings into question the sign,

timing, and magnitude of this feedback, which is a challenge for paleoclimate and future modeling. Considering all factors, it is possible that the Hami Basin was a significant dust-emitter at some point in the past when it was covered with fine-grained material (but maintained weaker winds) and may return to this state at some point in the future once the gravel cover is removed (but winds may return to weaker conditions as well, holding all else constant). It is imperative to consider not only the albedo effect of a changing land surface but also the characteristics of the erodible material if paleo-dust emissions are to be accurately modeled. The effect of land surface changes on dust emissions is a target for future work.

The concept advanced here is cautionary: the application of a modern view of surficial geology to paleoclimate models would have resulted in errors in the spring mean near-surface winds in the Turpan-Hami depression of up to ~25%, overestimating the frequency of dust-deflating wind speed hours, and unrecognized downwind effects because of local changes in albedo, sensible heat, and the planetary boundary layer. This region was chosen to model due to its high signal-to-noise ratio arising from its steep temperature gradients, and while we acknowledge that no synoptic-scale changes are evident within our largest experimental domain, other areas likely experience the same mechanism. For example, if such a boundary condition error was to be extrapolated to the low-albedo, rocky interior of Asia (Supplementary Fig. 1), its misrepresentation could negatively impact our understanding of past winds and aerosol production.

The desert and processes described here are not unique. Globally, windy, sediment-, and vegetation-starved desert landscapes can evolve to stone-armored surfaces if supported by the surface mantling bedrock. The wind-albedo-wind feedback is a global process, although potentially smaller than recognized here. Stony deserts in Antarctica[57,58], Death Valley[23], Australia[59,60], and Iran[61] are examples of areas that could have formed in part by the same wind–albedo–wind processes, the latter two acting as significant modern (and potentially paleo) dust sources. Finally, as recent work has pointed to this process acting as a primary driver of wind erosion on Mars[52], quantitatively describing the relationship between landscape evolution and surface atmospheric processes may have impacts for understanding processes on other planets within our solar system.

## Methods

**Model setup**. A one-way, tripled nested configuration of the regional WRF-ARW model version 3.9[62] was utilized to perform high-resolution simulations over the Hami Basin. The outer, middle, and inner domains had 30 , 10 , and 3.33 km horizontal grid spacings, respectively. The outer domain covered an area of 1800 km by 1500 km, while the middle and inner domains covered areas of 1180 km by 880 km and 403 km by 403 km, respectively (Supplementary Fig. 2). Time steps of 90, 30, and 10 s were used for each domain. The model atmosphere was decomposed into 60 vertical levels.

The Morrison 2-Moment microphysics scheme, which calculates the mixing ratio and number concentrations of five hydrometeor species (cloud, rain, ice, snow, and graupel), was used for cloud microphysics processes[63]. We also used the RRTMG LW and SW radiation schemes[64], the MYJ planetary boundary layer scheme[65], the Kain–Fritsch convection scheme in the two outer domains[66], the Monin–Obukhov similarity theory for the surface layer, and the Unified Noah Land Surface scheme[67]. Initial input data for land-based variables (i.e., vegetation cover, land-use, albedo, etc.) used for preprocessing of the domains described above is from the 24-category U.S. Geological Survey (USGS) data set[28,29]. The model was forced at the lateral boundaries (6-h intervals) and initialized with the NCEP Global Forecast System Analysis (GFS-ANL), which has spatial resolution of 0.5° × 0.5°. Model output was stored at hourly intervals for the two innermost domains, while that of the outer domain was stored at 3-h intervals. Advection was positive definite. The model was initialized at 00:00:00 UTC February 1, 2011, and run for the duration of the spring season until 10:00:00 UTC June 1, 2011. The specific period of interest considered for the run setup was the boreal spring (March, April, and May) because most dust storms occur in East Asia during this time[6,7,44]. February (00:00:00 UTC February 1, 2011—18:00:00 UTC February 28, 2011) was considered model spin-up and excluded from any analysis presented here. Year 2011 was used because it was not a year where Hami recorded the

hottest surface temperature[25]. Supplementary Table 4 provides a summary of all assigned model schemes.

To assess how changes in surface albedo over time may have impacted surface wind speeds, the background albedo variable of the 24-category USGS data set was altered for a box that roughly bounds the Hami Basin (Lat: 42.2°N-43.0°N; Lon: 91.1°E-93.8°E, outer domain). The albedo was set to 50, 40, 30, 20, and 10% for each simulation. The output from the 50% run is likely an overestimate of any effects, as most light-colored sandy deserts (such as the Sahara or Taklimakan Desert) have albedos of ~25–45%[28,29]. As geologic evidence points to the paleo-Hami Basin having similar surface features to these light-colored deserts, this paper focuses on temperature, pressure, and wind speed changes associated with the 40% simulation. The 10% variation of the run setup acts as a secondary control run, as this is approximately the albedo of the basin today.

**Regridding WRF Control output for model validation.** WRF_Control simulation output from Domain 1 was compared with GFS-ANL and ERA5[32] reanalysis data to validate the use of the WRF model in describing atmospheric processes over the western Gobi Desert, China, for the spring of 2011. WRF model output is on a curvilinear grid with ~30 -km grid spacing, while both global reanalysis data sets are provided on rectilinear grids at 0.5° × 0.5° resolution. To move beyond a simple visual comparison of specific atmospheric variables (including 2-m temperature, 10-m and 500-mb horizontal wind speeds, and surface pressure), it was necessary to place the various outputs onto a common grid system. To reduce the amount of interpolation, we chose to recalculate the higher-resolution WRF_Control output on the lower-resolution, rectilinear grid of both reanalysis data sets individually. To do this, we used the Universal Regridder for Geospatial Data module for Python (xESMF)[68], applying a bilinear interpolation to WRF_Control. This was repeated for all three months of interest (MAM). Monthly means of listed variables for both the model and reanalysis data sets were calculated by averaging all 3-h time steps within the month.

Daily average station observation data from the Global Surface Summary of the Day (GSOD), originally derived from The Integrated Surface Hourly (ISH) dataset (National Centers for Environmental Information (NCEI), 2018)[33], was used to asses WRF's ability to reproduce local surface conditions. For comparison with individual station observations, output from WRF_Control's Domain 2 was re-gridded, again using the bilinear interpolation from xESMF[68] onto a ~0.08° × 0.08° rectilinear grid. This spatial resolution was chosen because of its similarity to the original grid spacing of Domain 2, i.e., 10 km. Using the provided station locations[33], we calculated the average daily 2-m temperature (T2), surface pressure (SP), and 10-m horizontal wind speed (10-Winds) from WRF_Control for a 0.1° × 0.1° box centered on the station latitude and longitude.

**Statistical treatment.** To quantitatively compare the reanalysis and observational data sets with the WRF_Control simulation, mean bias error (MBE), mean absolute error (MAE), and RMSE for monthly averages at each grid point were calculated (Supplementary Tables 2, 3). The same was done for the daily averages computed in Domain 2 for comparison with the daily-averaged individual station observation data, with the addition of $r^2$ and $p$-values. $r^2$ values here indicate whether a relationship between the observation and model data exist, and $p$-values provide uncertainty constraints.

To understand the effects of albedo on surface and planetary boundary layer conditions, mean, medians, standard deviations, maximums, and minimums were calculated for all model simulations, where each value was determined from all grid points and time steps within the region where the albedo was altered (Supplementary Data 1). The variables considered were T2, HFX, 10-Winds, W, <~900-m averaged W (to represent a majority of the planetary boundary layer), TKE, <~900-m averaged TKE (to represent a majority of the planetary boundary layer), and PBLH.

For mean spring plots presented in Fig. 2a and b, the mean was calculated by averaging over all time steps of MAM for each grid point. The monthly means in Fig. 3a, b are calculated by averaging each time step and all grid points within the region where albedo was altered using Domain 3 output. The frequency plot (Fig. 4a) contains the hourly 10-Winds averaged over all grid points within the albedo box for the spring of 2011.

All percent differences presented in tables, figures, and discussed in the text are computed as relative differences from the control simulation.

## Data availability

W.R.F. simulation output required to evaluate and reproduce the calculations, figures, and tables found in this paper are available at figshare.com (https://doi.org/10.6084/m9.figshare.10013108). Full model output files will be provided by the author upon request.

## Code availability

Code used for calculations and generation of figures and tables will be provided by the author upon request.

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

## Acknowledgements

We would like to thank Sean Ridge, Mingfang Ting, Joerg Schaefer, and Arlene Fiore for fruitful discussions that improved the project. This research was funded in part by the U.S. National Science Foundation PIRE award #1545859 (Alex Pullen and Gisela Winckler), Clemson University, and Lamont-Doherty Earth Observatory.

## Author contributions

J.A., A.P. and Z.L. designed the project and developed the conceptual framework; A.P. and P.K. performed geological fieldwork; J.N. provided the observational data; J.A. and Z.L. performed the model simulations; J.A., Z.L. and L.G. analyzed the model output and performed calculations; J.A. and A.P. led the paper writing, with assistance from Z.L., G.W., P.K. and A.M.; all authors provided comments and revisions.

## Competing interests

The authors declare no competing interests.
