## [Peer Review File · Nature Communications]

Reviewers' comments:

Reviewer #1 (Remarks to the Author):

What are the major claims of the paper?

The manuscript proposes and investigates the hypothesis that wind changes the surface albedo in arid regions as fine sediments are eroded by wind, which in turn will reduce the albedo as the remaining coarse sediment (e.g. stones) may have a darker color, which in turn will change the thermal properties of the surface. In topographic basins (bright surface; higher albedo) surrounded by (darker) mountains (lower albedo), the reduced albedo of the basin's bottom sediments may reduce the difference in albedo between mountain range and basin. As the albedo relates to the amount of energy a surface absorbs, a spatial gradient in albedo can be referred to a thermal gradient, which in turn results into a local pressure gradient consequently modulating the regional wind fields. A reduced albedo gradient may lead to lower surface wind speeds for cases where the albedo gradient enforces the local wind.

To test the hypothesis, the authors used the atmosphere model WRF over part of the western Gobi Desert. As the Gobi Desert is characterized by stony surfaces, this land surface type may have undergone an evolution fostered by erosion ultimately lowering the albedo at millennial time scale. To test for the albedo impact solely, the albedo is the only parameter that was changed.

The manuscript is clearly structured and the study is well motivated. The data description is technically sound and the authors provide sufficient information on model setup. The model data itself supports the hypothesis of the wind-albedo-wind feedback and proves the concept - which can be expected as the WRF model is designed to do so. However, as any model validation (e.g. against observational data sets) is missing, it remains open how well the reference model setup represents the actual conditions.

Are the claims novel?

The individual components of the hypothesis are known and the acknowledged by the authors. The authors rather suggest, that these known albedo feedbacks are not accurately included in climate models and therefore contribute to uncertainties in wind speed reconstructions for ancient climates.

In order to underline the importance of accounting for landscape evolution by (wind) erosion, I wish the authors could provide more information on the significance of the change in wind speed due to change in albedo. As the results are shown for a rather small area and, and no interactions on e.g. synoptic scale circulation are discussed, the significance remains somewhat limited to a proof of a concept that is known. Furthermore, having some information on the performance of the model respective how realistic the wind speed distribution over the Tarim Basin is represented may help to illustrate the significance.

Will the paper be of interest to others in the field? Will the paper influence thinking in the field?

The paper underlines the importance of an up-to-date surface albedo implemented in atmosphere models. Whereas model simulations covering the satellite era can be updated by satellite retrieved albedo maps, albedo maps used for the past are calculated from e.g. vegetation models accounting for a change in vegetation type and cover. Change in soil conditions e.g. due to erosion are less regarded. Hence, the authors point towards an important issue. However, it feels the main arguments are only briefly brought up and not as thoroughly discussed as could be. Maybe the final paragraph may be used to provide a focused conclusion rather than a general statement.

I therefore wish the authors to strengthen their arguments by e.g. illustrating why the differences in wind speed matter. To someone not into paleo-wind reconstructions, what are the (quantitative) consequences of a monthly averaged change in wind speed of 0.75 m/s (Fig. 3) e.g. for the total dust emission flux? Maybe you can find a clever way to illustrate how much it matters? You may want consider including variables other than wind speed too. E.g. (potential) temperature, turbulent kinetic energy (TKE), and the Bowen ratio reflecting the characteristics of the planetary boundary layer determining the gustiness. Furthermore, how strongly are synoptic scale features affected by local scale albedo changes? This may contribute to the general discussion on the

impact as well as regional atmospheric circulation and local wind systems are superimposed.

Reviewer #2 (Remarks to the Author):

The paper presents a feedback mechanism between thermal-driven winds and surface sediment conditions, whereby the development of surface armouring from wind-driven soil-erosion leads to more intense thermal-driven winds. The paper presents weather model simulations using the different surface conditions to show the impact on wind patterns and strengths.

The wind simulations are very thorough and insightful, and the novelty claim lies in the bedrock-atmosphere feedback (as per line 152).

The crux of the hypothesis lies in the process of the surface armouring to develop. The armouring process is described in general in line 77-78, and the site-specific history in lines 89-93. On the general process the paper cites four sources, three of which are severely outdated (from 1904, 1968, and 1970) and the fourth reflects a local study. Current understanding of desert pavements suggests that they are actively maintained and indeed accumulate underneath them aeolian silt and dust deposits. This contemporary understanding started with the work by McFadden et al. (1987) and is reflected in later work by, for example, Pelletier et al. (2007). The previous site-specific study by the authors cited here (Pullen et al., 2018) presents a clearer picture of the development of the surface armouring, suggesting that wind erosion has removed layers of playa lake (silt and clay) sediments until interlaced conglomerate layers are exposed and no further (aeolian) erosion is possible.

On the whole this hypothesis is reasonable and the consequences of the changes in surface conditions over time in terms of thermally-driven wind strength appears logical.

The paper presents credible evidence that the bedrock-atmosphere feedback mechanism has significant impacts on paleo-climate investigations and dust source and sink identification, both in the specific site studied here as well as elsewhere around the globe.

I recommend that the paper is published, provided that the authors update the description and literature on surface armouring and desert pavements to the contemporary understanding, and that the authors reflect in the text on the possible role of the local surface armouring acting as a potential dust trap.

McFadden LD, Wells SG, Jercinovich MJ. 1987. Influences of eolian and pedogenic processes on the origin and evolution of desert pavements. *Geology* 15: 504–508.

Pelletier, J.D., Cline, M. and DeLong, S.B., 2007. Desert pavement dynamics: numerical modeling and field-based calibration. *Earth Surface Processes and Landforms: The Journal of the British Geomorphological Research Group*, 32(13), pp.1913-1927.

Reviewer #3 (Remarks to the Author):

This research presents a compelling hypothesis that the formation of gravel-mantled surfaces reduces the albedo of desert surfaces, thus resulting in an increase in wind speeds. To my knowledge, this hypothesis has not been previously considered in the literature and represents a novel advancement in our knowledge of landscape evolution. This hypothesis is supported by fieldwork and observations that deflation results in the formation of extensive, densely-packed gravel surfaces (Pullen et al., 2018) and modeling that isolates one variable – albedo – to demonstrate how wind speed increases with a decrease in albedo that resulted from the formation of gravel-covered surfaces (results presented in this paper). This result is not surprising; however, this is a unique finding with broad implications for modeling wind regimes past and present. As the authors note, this wind-albedo feedback extends beyond the area of gravel cover, potentially impacting large desert areas. While the gravel surfaces themselves become less important dust sources over time, the other landforms and soils adjacent to these gravel surfaces may be susceptible to wind erosion as a result of this feedback. I find this to be a significant advancement in how we think about landscape dynamics, dust emissions, and feedbacks in deserts. The article is well-written. I would like to pose two questions for the authors to consider:

1. As gravel surfaces coalesce into pavements, their surface roughness can decrease (for example,

Frankel and Dolan, 2007, Characterizing arid region alluvial fan surface roughness with airborne laser swath mapping digital topographic data, JGR-ES v. 112), thus also potentially resulting in increased wind speeds at the surface. Surface roughness plays a considerable role in surface wind speed (and dust emission), and is only mentioned briefly in this study.

2. As gravel surfaces mature, they darken by accumulating desert varnish, significantly reducing albedo. Are the gravel surfaces in the Hami varnished, or natural rock color? The oldest gravel surfaces with well-developed varnish may have the lowest albedo, whereas younger gravel surfaces with weak varnish may have slightly higher albedo.

How might these two variables also contribute to increasing wind speeds over time? It seems there may be multiple feedbacks that may combine to result in increased wind speeds over time. The basic concept as presented (albedo-wind feedback) is significant, but I think the overall story could be improved by acknowledging these two other important and related variables.

Mark Sweeney

We would like to thank all three reviewers for providing constructive and thoughtful feedback that led to significant improvements in the manuscript.

There are three main changes made to the manuscript: 1) the addition of a quantitative description of WRF model performance against observations, which can be found in the “Results” and “Methods” sections; 2) new results for additional variables, including vertical wind speed and turbulent kinetic energy at the surface and in the upper 900m, the height of the planetary boundary layer, sensible heat flux, and 2-meter temperature, found in the “Results” section; and 3) an expanded discussion on the implications and uncertainty of attempting to reconstruct winds and dust in the past, present, and future when this feedback system is not considered.

Below, we address every reviewer comment line-by-line below. Reviewer comments are displayed in **bold**, and our responses are in non-bolded **red font**.

Reviewer #1:

The manuscript proposes and investigates the hypothesis that wind changes the surface albedo in arid regions as fine sediments are eroded by wind, which in turn will reduce the albedo as the remaining coarse sediment (e.g. stones) may have a darker color, which in turn will change the thermal properties of the surface. In topographic basins (bright surface; higher albedo) surrounded by (darker) mountains (lower albedo), the reduced albedo of the basin's bottom sediments may reduce the difference in albedo between mountain range and basin. As the albedo relates to the amount of energy a surface absorbs, a spatial gradient in albedo can be referred to a thermal gradient, which in turn results into a local pressure gradient consequently modulating the regional wind fields. A reduced albedo gradient may lead to lower surface wind speeds for cases where the albedo gradient enforces the local wind.

To test the hypothesis, the authors used the atmosphere model WRF over part of the western Gobi Desert. As the Gobi Desert is characterized by stony surfaces, this land surface type may have undergone an evolution fostered by erosion ultimately lowering the albedo at millennial time scale. To test for the albedo impact solely, the albedo is the only parameter that was changed.

The manuscript is clearly structured and the study is well motivated. The data description

is technically sound and the authors provide sufficient information on model setup. The model data itself supports the hypothesis of the wind-albedo-wind feedback and proofs the concept - which can be expected as the WRF model is designed to do so. However, as any model validation (e.g. against observational data sets) is missing, it remains open how well the reference model setup represents the actual conditions.

Excellent point, and we appreciate the reviewer's positive comments on the overall manuscript and its main points. While the Weather Research and Forecasting (WRF) model is one of the most used regional atmospheric models, we agree that model validation was lacking. To that end, we have validated the model in the following ways:

1) We quantitatively described how well our control simulation (WRF_Control) reproduced 2-m temperature (T2), surface pressure (SP), 500-mb horizontal winds (500-Winds), and 10-m horizontal winds (10-Winds) from two global reanalysis datasets (GFS-ANL at 0.5° x 0.5° resolution and ECMWF-ERA5 at 0.5° x 0.5°) for the months of March, April, and May of 2011. A description of the reanalysis datasets is provided in the "Results" and "Methods" section of the manuscript (see lines 91-105 and 333-356).

WRF_Control Domain 1 results from the variables listed above were first re-gridded (mapped onto the same spatial resolution and rectilinear coordinate system) as the ERA5 and GFS-ANL datasets (a more detailed description is provided in the "Methods" section (lines 334-356). This allows us to go beyond a qualitative comparison (visually assessing the differences between model output and reanalysis data). Three different statistical quantities (mean bias error, mean absolute error, and root-mean-squared error) are calculated for each month and provided in Table S2). In addition, we do provide plots of all variables and months previously mentioned for those that prefer a visual comparison (Fig. S3-S4).

As evident from the both the quantitative and qualitative comparisons, WRF_Control accurately reproduces T2, SP, 500-Winds, and 10-Winds for all months when compared to the GFS-ANL output. As mentioned in the manuscript (lines 101-105 and supplement lines 14-19), while it is important that our WRF_Control simulation can reproduce the reanalysis dataset that is used to force the model at its boundaries (GFS-ANL), it is crucial that it also be compared to an independent set of observations, which in this case is the ERA5 reanalysis dataset. Again, WRF_Control does well in representing the regional climate.

We note that the model does show a positive bias for 10-Winds, averaging ~2 m/s when compared to both reanalysis datasets (Fig. S4). The reasoning for this can now be found in the supplement (lines 14-34). To provide a quick summary here for the reviewers, we suggest that there may be two factors which can lead to this biasing of 10-Winds. One has to do with the higher resolution WRF simulation incorporating faster wind speeds near steep topography. Even when the product is mapped onto the grids of the two lower resolution reanalysis datasets, there is some incorporation of higher wind speeds through the bilinear interpolation method (see lines 334-356 in the manuscript for a more thorough description of this process). Higher wind speeds produced in the WRF model has been previously documented, in even higher resolution simulations (Jimenez et al., 2013a/2013b). This does not always mean that the WRF model is incorrect in producing near-surface wind speeds. See lines 26-34 in the supplement for a

discussion of a location near Urumqi, Xinjiang, China, where higher wind speeds (~6 m/s averaged annually) are observed from ground-based measurements and are reproduced by WRF but are not seen in the two reanalysis datasets. A second cause of the faster winds at the surface may stem from the feedback between the two nested inner domains (Domains 2 and 3) and the outer domain (Domain 1) in our simulations. In either case, overall, WRF does replicate regional surface temperature, pressure, winds, and winds aloft to a high degree.

2) We quantitatively described how well our control simulation (WRF_Control) reproduces T2, SP, and 10-Winds from ground-based observations for the spring of 2011. A description of the observational datasets is provided in the “Results” and “Methods” section of the manuscript (see lines 91-105 and 333-356).

While we assess the ability of WRF_Control to accurately reproduce regional climate for the spring of 2011 using global, lower-resolution reanalysis datasets, we wanted to take this one step further and attempt to validate WRF’s skill at the local (city) level. To do this, we compare output of T2, SP, and 10-Winds from WRF_Control’s Domain 2 to hourly observations from 14 sites found throughout western China (Fig. S6). This also allows us to better study and understand the bias in 10-Winds, as some sites lie near steep topography.

WRF_Control Domain 2 results from the variables listed above were first re-gridded onto a rectilinear grid at the same resolution (~0.08° x 0.08°) as the original simulation. In this way, no resolution is lost, but a more accurate comparison can be made given the latitude and longitudes listed for the observational station (see lines 334-356 in manuscript for details). For all but one of the sites (Balguntay), our WRF_Control simulation accurately reproduces T2 and SP (see Table S2). In addition, excluding the sites of Balguntay and Baytik Shan, our control run does well in representing 10-Winds. Specifically, and in our view critically, WRF_Control is able to reproduce 10-meter horizontal wind speeds at Shisanjianfang to a high degree ($r^2=0.80$ and RMSE=2.41, where mean winds are ~10 m/s and reach ~30 m/s), which lies directly in the region where we alter albedo to test our hypotheses.

Again, there is a positive bias for some of the sites, which may be due to averaging over an area that includes higher near-surface winds from steep topography. However, overall, the model does well in reproducing local conditions for the spring of 2011.

We hope that this thorough and quantitative analysis of the WRF_Control simulation against reanalysis and ground-based observations serves in validating its use to test our proposed feedback system.

The individual components of the hypothesis are known and the acknowledged by the authors. The authors rather suggest, that these known albedo feedbacks are not accurately included in climate models and therefore contribute to uncertainties in wind speed reconstructions for ancient climates.

We thank the reviewer for pointing out one of our key conclusions. We would like to make clear that while we acknowledge that the physics behind the feedbacks are known, the feedback system we describe (a wind-albedo-wind feedback driven by landscape evolution) has not previously, to our knowledge, been quantitatively examined in the literature.

In order to underline the importance of accounting for landscape evolution by (wind) erosion, I wish the authors could provide more information on the significance of the change in wind speed due to change in albedo. As the results are shown for a rather small area and, and no interactions on e.g. synoptic scale circulation are discussed, the significance remains somewhat limited to a proof a concept that is known. Furthermore, having some information on the performance of the model respective how realistic the wind speed distribution over the Tarim Basin is represented may help to illustrate the significance.

Again, an excellent point. We have added the following discussion (see below and lines 129-183 and 221-231 in the manuscript) and figures (Fig. 2, Figs. S4) to highlight the significance of the change in wind speed (and associated variables) resulting from the changes in albedo.

1) We have updated our “Results” section so that now output for variables important to the feedback system we discuss and the 10-m winds originally presented, such as 2-m temperature, sensible heat flux, surface and 900-m averaged vertical wind speeds, surface and 900-m averaged turbulent kinetic energy, and the height of the boundary layer, is reported. However, we do not delve too deeply into the implications that changes in these other variables may have, as this will be more closely examined in a follow-up dust modeling study.

2) We have expanded our discussion on the regional effects observed in our simulations from altered albedos within the Hami Basin. See lines 221-231.

3) We have further discussed the impacts of albedo on absolute mean near-surface wind speeds and the frequency of high wind events (gustiness). In our opinion, this is where this previously unrecognized wind-albedo-wind feedback is most critical, as it could alter dust emissions and erosion to a much larger degree than the 25% change in wind speed, as dust flux is related to the cube of near-surface horizontal wind speed (see text for references).

4) While any dust calculations or modeling experiments are outside the scope of this manuscript, we did add an extensive discussion on the compounding factors that landscape evolution via wind erosion in the Hami Basin could cause and how these may play into dust emissions. One obvious conclusion we draw is that the increased frequency of hours spent at high wind speeds during periods of low albedo (similar to present conditions) and less hours at high albedos (past conditions) would effect dust emissions significantly (at least the ~40% observed in the frequency of winds in the 4-12 m/s range). This is actually an important driver of the “paradox” component of this work; the region was once full of fine-grained, friable material but experienced weaker winds and less gusty conditions. As time progressed, this easily erodible material became less abundant, but winds picked up. Thus, at some point in the past (what can be thought of as a geologic and climatic sweetspot), the Hami Basin was likely a large emitter of dust.

5) We do not discuss synoptic-scale impacts because they do not exist, or at least are not recognized for the parameters important to our focus here. The reason that we chose to model the effects of changing surface albedo in the Hami Basin is because of the likely large signal-to-noise ratio, stemming from the high wind speeds, steep topography, and extreme temperature gradients of the system. However, because of the relatively small scale of the basin and experiment in general, synoptic-scale impacts are not to be expected, particularly for distant upper-level winds, pressure, and temperatures. This is not to say that the associated dust emissions from the Hami Basin that are proposed in the discussion of the manuscript (and will be studied in detail in a follow-up paper) could not have far-ranging effects, particularly for ocean biogeochemistry or radiative properties of the surface. We leave the discussion of synoptic scale effects on climate for our subsequent studies on 1) Hami dust production with altered surface characteristics and winds and 2) the expansion of this wind-albedo-wind feedback system to the larger Gobi Desert.

6) We discuss in the previous responses the addition of model validation against ground-based observation and reanalysis datasets. We provide quantitative comparisons for a large region (Domain 1) that contains the Tarim Basin specifically as well as two observational sites that fall within the Tarim Basin.

The paper underlines the importance of an up-to-date surface albedo implemented in atmosphere models. Whereas model simulations covering the satellite era can be updated by satellite retrieved albedo maps, albedo maps used for the past are calculated from e.g. vegetation models accounting for a change in vegetation type and cover. Change in soil conditions e.g. due to erosion are less regarded. Hence, the authors point towards an important issue. However, it feels the main arguments are only briefly brought up and not as thoroughly discussed as could be. Maybe the final paragraph may be used to provide a focused conclusion rather than a general statement.

In the interest of brevity, a robust discussion was lacking. We have improved the discussion through the points presented in the response directly above. Again, this can be observed in the new text (lines 232-272).

I therefore wish the authors to strengthen their arguments by e.g. illustrating why the differences in wind speed matter. To someone not into paleo-wind reconstructions, what are the (quantitative) consequences of a monthly averaged change in wind speed of 0.75 m/s (Fig. 3) e.g. for the total dust emission flux? Maybe you can find a clever way to illustrate how much it matters? You may want consider including variables other than wind speed too. E.g. (potential) temperature, turbulent kinetic energy (TKE), and the Bowen ratio reflecting the characteristics of the planetary boundary layer determining the gustiness. Furthermore, how strongly are synoptic scale features affected by local scale albedo changes? This may contribute to the general discussion on the impact as well as regional atmospheric circulation and local wind systems are superimposed.

We have addressed these points in previous responses above.

Reviewer #2:

The paper presents a feedback mechanism between thermal-driven winds and surface sediment conditions, whereby the development of surface armouring from wind-driven soil-erosion leads to more intense thermal-driven winds. The paper presents weather model simulations using the different surface conditions to show the impact on wind patterns and strengths.

The wind simulations are very thorough and insightful, and the novelty claim lies in the bedrock-atmosphere feedback (as per line 152).

We appreciate the reviewer's positive comments on the overall manuscript and its main points.

The crux of the hypothesis lies in the process of the surface armouring to develop. The armouring process is described in general in line 77-78, and the site-specific history in lines 89-93. On the general process the paper cites four sources, three of which are severely outdated (from 1904, 1968, and 1970) and the fourth reflects a local study. Current understanding of desert pavements suggests that they are actively maintained and indeed accumulate underneath them aeolian silt and dust deposits. This contemporary understanding started with the work by McFadden et al. (1987) and is reflected in later work by, for example, Pelletier et al. (2007). The previous site-specific study by the authors cited here (Pullen et al., 2018) presents a clearer picture of the development of the surface armouring, suggesting that wind erosion has removed layers of playa lake (silt and clay) sediments until interlaced conglomerate layers are exposed and no further (aeolian) erosion is possible.

We thank the reviewer for pointing out updated and more relevant citations for our discussion on the surface armorings process and the development of the topographically-tiered unconsolidated gravel deposits responsible for the albedo changes in the Hami Basin. We have updated this section (lines 63 to 71) to reflect the reviewer's suggestions.

On the whole this hypothesis is reasonable and the consequences of the changes in surface conditions over time in terms of thermally-driven wind strength appears logical.

We thank the reviewer for their positive comments.

The paper presents credible evidence that the bedrock-atmosphere feedback mechanism has significant impacts on paleo-climate investigations and dust source and sink identification, both in the specific site studied here as well as elsewhere around the globe.

We thank the reviewer for their positive comments.

I recommend that the paper is published, provided that the authors update the description and literature on surface armouring and desert pavements to the contemporary understanding, and that the authors reflect in the text on the possible role of the local surface armouring acting as a potential dust trap.

We have made the reviewer's suggested alterations. See lines 63 to 71 on surface armorings, and lines 232-256 for the effects of armorings on dust trapping.

McFadden LD, Wells SG, Jercinovich MJ. 1987. Influences of eolian and pedogenic processes on the origin and evolution of desert pavements. *Geology* 15: 504–508.
Pelletier, J.D., Cline, M. and DeLong, S.B., 2007. Desert pavement dynamics: numerical modeling and field-based calibration. *Earth Surface Processes and Landforms: The Journal of the British Geomorphological Research Group*, 32(13), pp.1913-1927.

These references have been added, as have two others (Wells et al., 1995, and Laity et al., 2011).

Reviewer #3:

This research presents a compelling hypothesis that the formation of gravel-mantled surfaces reduces the albedo of desert surfaces, thus resulting in an increase in wind speeds. To my knowledge, this hypothesis has not been previously considered in the literature and represents a novel advancement in our knowledge of landscape evolution. This hypothesis is supported by fieldwork and observations that deflation results in the formation of extensive, densely-packed gravel surfaces (Pullen et al., 2018) and modeling that isolates one variable – albedo – to demonstrate how wind speed increases with a decrease in albedo that resulted from the formation of gravel-covered surfaces (results presented in this paper). This result is not surprising; however, this is a unique finding with broad implications for modeling wind regimes past and present. As the authors note, this wind-albedo feedback extends beyond the area of gravel cover, potentially impacting large desert areas. While the gravel surfaces themselves become less important dust sources over time, the other landforms and soils adjacent to these gravel surfaces may be susceptible to wind erosion as a result of this feedback. I find this to be a significant advancement in how we think about landscape dynamics, dust emissions, and feedbacks in deserts. The article is well-written.

We appreciate the reviewer's positive comments on the overall manuscript and its main points.

I would like to pose two questions for the authors to consider:

1. As gravel surfaces coalesce into pavements, their surface roughness can decrease (for example, Frankel and Dolan, 2007, Characterizing arid region alluvial fan surface roughness with airborne laser swath mapping digital topographic data, *JGR-ES* v. 112), thus also potentially resulting in increased wind speeds at the surface. Surface roughness

plays a considerable role in surface wind speed (and dust emission), and is only mentioned briefly in this study.

We agree that this is a point that should be considered. We have lines 232-256 to reflect this point.

2. As gravel surfaces mature, they darken by accumulating desert varnish, significantly reducing albedo. Are the gravel surfaces in the Hami varnished, or natural rock color? The oldest gravel surfaces with well-developed varnish may have the lowest albedo, whereas younger gravel surfaces with weak varnish may have slightly higher albedo.

We have added some text to introduce this idea to the reader (lines 249 to 256). Some of the unconsolidated gravels in Hami have developed desert varnish. However, the distribution of desert varnished stones in the Hami is heterogenous and significantly less abundant than stones showing ventifact or more moderately weathered surfaces that closely approximate the rock's natural color. Fortunately for this investigation the cobbles and pebbles that compose the unconsolidated gravels are naturally dark in color thus the question of the spatial-temporal distribution of desert varnish and its effects on albedo and thus wind speed is muted. Pullen et al., (2018) Figure 7 shows photos of the unconsolidated gravels in Hami. The photos indicate the predominance of naturally dark colored stones and minimal desert varnish. The natural colors of the gravels are dark owing to the mostly dark colored Paleozoic–Mesozoic rocks in the eastern Tian Shan for which the cobbles and pebbles were likely sourced prior to being deposits in the late Cenozoic (alluvial fan) strata (described here) that were then deflated by the wind. But we agree wholeheartedly with the reviewer that if the gravels were predominately composed of cobbles and pebble of naturally light colors, the spatial-temporal variation in desert varnish would be massively important to the spatial-temporal change in the albedo of the basin. We note that this idea should be fully considered if the lessons learned in Hami are to be accurately exported.

How might these two variables also contribute to increasing wind speeds over time? It seems there may be multiple feedbacks that may combine to result in increased wind speeds over time. The basic concept as presented (albedo-wind feedback) is significant, but I think the overall story could be improved by acknowledging these two other important and related variables.

See responses provided above.

REVIEWERS' COMMENTS:

Reviewer #2 (Remarks to the Author):

I am satisfied with the revisions the authors have made in response to my review and I am happy to recommend this manuscript for publication.

Reviewer #3 (Remarks to the Author):

This is my second review of this manuscript. I have read through the reviewer comments and replies to comments by the authors, as well as the revised manuscript. 1) The authors have adequately dealt with my comments, and I appreciate that they considered my suggestions. 2) I found that the comments and revisions pertaining to the WRF model and validation to be helpful in providing additional context and clarity with regards to the modeling.3) Some additional references and text modifications clarify their use of the term "desert pavement".

The manuscript is still clear to read and much improved. I have no additional comments or suggestions. It presents a compelling hypothesis tested with the WRF model that I hope the editors find acceptable to publish in Nature.

Mark Sweeney

We would like to thank the two reviewers and the editor for providing constructive and thoughtful feedback on the revised version of this manuscript.

Below, we address reviewer comments that are not shown via track changes in the word document. Reviewer comments are displayed in **bold**, and our responses are in non-bolded red font.

Reviewer #2:

I am satisfied with the revisions the authors have made in response to my review and I am happy to recommend this manuscript for publication.

We thank the reviewer for their positive comments.

Reviewer #3:

This is my second review of this manuscript. I have read through the reviewer comments and replies to comments by the authors, as well as the revised manuscript. 1) The authors have adequately dealt with my comments, and I appreciate that they considered my suggestions. 2) I found that the comments and revisions pertaining to the WRF model and validation to be helpful in providing additional context and clarity with regards to the modeling.3) Some additional references and text modifications clarify their use of the term "desert pavement".

The manuscript is still clear to read and much improved. I have no additional comments or suggestions. It presents a compelling hypothesis tested with the WRF model that I hope the editors find acceptable to publish in Nature.

We thank the reviewer for their positive comments.